# The Impact of Mind-Body Exercises on Motor Function, Depressive Symptoms, and Quality of Life in Parkinson’s Disease: A Systematic Review and Meta-Analysis

**DOI:** 10.3390/ijerph17010031

**Published:** 2019-12-18

**Authors:** Xiaohu Jin, Lin Wang, Shijie Liu, Lin Zhu, Paul Dinneen Loprinzi, Xin Fan

**Affiliations:** 1Department of Physical Education, Wuhan University of Technology, Wuhan 430070, China; jxh126pd@126.com; 2School of Physical Education & Training, Shanghai University of Sport, Shanghai 200438, China; 1911111017@sus.edu.cn; 3School of Physical Education, Soochow University, Suzhou, Jiangsu 205301, China; zhulin999999999@126.com; 4Exercise & Memory Laboratory, Department of Health, Exercise Science and Recreation Management, The University of Mississippi, Oxford, MS 38677, USA; pdloprin@olemiss.edu; 5College of Physical Education, Hubei Normal University, Huangshi 435002, China; fan_x163@163.com

**Keywords:** meta-analysis, Tai Chi, Parkinson disease, Yoga, Health Qigong, quality of life

## Abstract

**Purpose:** To systematically evaluate the effects of mind-body exercises (Tai Chi, Yoga, and Health Qigong) on motor function (UPDRS, Timed-Up-and-Go, Balance), depressive symptoms, and quality of life (QoL) of Parkinson’s patients (PD). **Methods:** Through computer system search and manual retrieval, PubMed, Web of Science, The Cochrane Library, CNKI, Wanfang Database, and CQVIP were used. Articles were retrieved up to the published date of June 30, 2019. Following the Cochrane Collaboration System Evaluation Manual (version 5.1.0), two researchers independently evaluated the quality and bias risk of each article, including 22 evaluated articles. The Pedro quality score of 6 points or more was found for 86% (19/22) of these studies, of which 21 were randomized controlled trials with a total of 1199 subjects; and the trial intervention time ranged from 4 to 24 weeks. Interventions in the control group included no-intervention controls, placebo, waiting-lists, routine care, and non-sports controls. Meta-analysis was performed on the literature using RevMan 5.3 statistical software, and heterogeneity analysis was performed using Stata 14.0 software. **Results:** (1) Mind-body exercises significantly improved motor function in PD patients, including UPDRS (SMD = −0.61, *p* < 0.001), TUG (SMD = −1.47, *p* < 0.001) and balance function (SMD = 0.79, *p* < 0.001). (2) Mind-body exercises also had significant effects on depression (SMD = −1.61, *p* = 0.002) and QoL (SMD = 0.66, *p* < 0.001). (3) Among the indicators, UPDRS (*I*^2^ = 81%) and depression (*I*^2^ = 91%) had higher heterogeneity; according to the results of the separate combined effect sizes of TUG (*I*^2^ = 29%), Balance (*I*^2^ = 16%) and QoL (*I*^2^ = 35%), it shows that the heterogeneity is small; (4) After meta-regression analysis of the age limit and other possible confounding factors, further subgroup analysis showed that the reason for the heterogeneity of UPDRS motor function may be related to the sex of PD patients and severity of the disease; the outcome of depression was heterogeneous. The reason for this may be the use of specific drugs in the experiment and the duration of intervention in the trial. **Conclusion:** (1) Mind-body exercises were found to have significant improvements in motor function, depressive symptoms, and quality of life in patients with Parkinson’s disease, and can be used as an effective method for clinical exercise intervention in PD patients. (2) Future clinical intervention programs for PD patients need to fully consider specific factors such as gender, severity of disease, specific drug use, and intervention cycle to effectively control heterogeneity factors, so that the clinical exercise intervention program for PD patients is objective, scientific, and effective.

## 1. Introduction

Parkinson’s disease (PD), also known as Idiopathic Parkinson’s disease, is one of the most common progressive neurodegenerative diseases that occur in the elderly [1]. With the degeneration of dopaminergic neurons in the substantia nigra and the formation of Lewy bodies as pathological features [2], patients have gait and posture disorders, resting tremor, bradykinesia, muscle stiffness, and other characteristic motor symptoms [3]; accompanied by non-motor disorders such as sleep disorders, cognitive decline, fatigue, anxiety, and depression [4]. PD can be treated clinically by levodopa. As the disease progresses and drug efficacy decreases, movement is aggravated by a series of complications. Depression is the most common comorbidity of PD [5]; studies have confirmed drugs cannot solve non-motor conditions such as depression and anxiety [6,7]. The side-effects caused by long-term medication can also seriously endanger health. These symptoms may not only cause a cognitive decline and loss of interest in life, but also may cause loss of independence, increased falls, and even death, leading to a reduced quality of life (QoL) [8,9]. To date, the etiology and pathogenesis of PD are unclear, and genetic factors, environmental toxins, and aging may all contribute to the disease. As the population ages, it is expected that the global prevalence of Parkinson will double by 2050 [10].

Based on the above factors, it is imperative to find other effective PD treatment programs. Although the occurrence of neurodegenerative changes cannot be prevented, exercise can be used to assist in rehabilitation of PD patients. Studies have reported that dance [11], resistance training [12], and stretching exercises [13] are effective in improving the function of patients with Parkinson’s disease. Mind-body exercise is a low-cost, easy-to-operate, low-impact, moderate- intensity aerobic exercise that emphasizes both skeletal muscle stretching and relaxation, physical coordination training, and emphasizes on breathing and movement control. The perfect combination of body and mind can have a positive effect on the body and mind, suitable for the rehabilitation of chronic diseases [14,15], especially for non-motor symptoms (depression, stress, pain, cardiovascular disease, and high blood pressure) [16,17,18]. At present, the common forms of physical and mental exercises mainly include Health Qigong, Tai chi, and Yoga. They combine the guidance of traditional culture, the technique of voluntarily-induced vomiting, and some basic theories of Chinese medicine (meridian theory and a series of self-cultivation methods), and thus treat PD symptoms with great efficacy [19,20].

In recent years, there have been more and more studies on the effect of physical and mental exercises on PD. Various research interventions have focused on Tai Chi, Yoga, and Health Qigong and their influence on PD motor function, depressive symptoms, and quality of life with [21,22,23,24,25,26,27,28,29,30,31,32,33,34,35,36,37,38,39,40,41,42,43,44,45,46,47]. This study, for the first time, combines these three forms of mind-body exercise and quantifies their effects to assess safety and potential benefits, in order to provide scientific evidence for their clinical use in improving motor function, depressive symptoms, and quality of life in patients with Parkinson’s Disease.

## 2. Materials and Methods

### 2.1. Literature Search

The following electronic databases were manually searched for specific terms: PubMed, Web of Science, The Cochrane Library, CNKI, Wanfang Database, and VIP Database (CQVIP). The search deadline was 30 June 2019. The following search terms were used in different combinations: Parkinson’s Disease, physical and mental exercise, Tai Chi/Tai ji, Yoga, Qigong/Health Qigong (Ba Duan Jin, Wu Qin Xi, Yi Jin Jing, Liu Zi Jue). To obtain additional literature and data, we also sought help by email and downloaded the data from the Appendix A source site.

### 2.2. Study Selection

The inclusion criteria were as follows: (1) randomized controlled trial (RCT) and controlled clinical trial (CCT); (2) target population diagnosed as PD; (3) type of intervention: experimental group included in the form of mind-body exercises (Tai Chi, Yoga, Health Qigong), compared with different types of control groups (i.e., no-intervention control group, placebo, waiting-list, routine care, and non-sports control group); (4) outcome indicators include test data on motor function, depressive symptoms, and quality of life; (5) published in Chinese or English; (6) subject age over 40 years. Experimental intervention time was four weeks or more.

Conditions excluded from the study were: (1) non-clinical controlled trials; (2) descriptive literature; (3) republished literature; (4) literature that did not include full text (e.g., abstracts); (5) studies that did not have relevant outcomes; after exclusion, the two authors independently determined eligibility based on the Cochrane Collaboration System Evaluation Manual (version 5.1.0). Differences were resolved by consensus after discussion. If there were different opinions, consultation with a third author ensured accuracy of the data extraction.

### 2.3. Data Abstraction

For eligible studies, two researchers independently screened the title, abstract, and full text, according to predetermined criteria. The extracted data included general information (first author, country and publication year, sample size, dropout rate, age range tested), details of the intervention (time of treatment duration, method, dose of intervention, and comparative details), and outcome indicators. Disputes were resolved by consensus after discussion.

### 2.4. Methodological Quality Assessment

Two researchers independently assessed the methodological quality of eligible studies using the Physical Therapy Evidence Database (PEDro) scale. The PEDro scale consists of 11 items: eligibility criteria, random assignment, covert assignment, baseline similarity group, subject blindness, therapist blindness, assessor blindness, under 15% dropout, intentional treatment analysis, inter-group statistical comparison, point measurement and variable analysis. The scale has revealed good reliability for the systematic evaluation of clinical randomized controlled trials. The higher the evaluation score, the better the quality of the method.

### 2.5. Statistical Analysis

Meta-analysis was performed using Cochrane collaboration software (Review Manager 5.3). Q value and *I*^2^ statistics were used to assess heterogeneity for variation in true effect sizes across the included studies. When there is statistical heterogeneity between the results (heterogeneity test *I*^2^ ≥ 50%, *p* < 0.10), the random effect model is used for meta-analysis. Otherwise, the fixed-effect model is used, and the standardized average is calculated. Difference (SMD) and 95% confidence interval (CI) were calculated. When *I*^2^ is greater than 75%, the heterogeneity is considered to be high, and sensitivity analysis is performed using a single study with a one-by-one exclusion method. Stata14.0 was used to identify the effect size of the study intervention as the dependent variable, which may affect meta-analysis heterogeneity factors (publishing years, sample size, intervention test period, etc.) as covariates, using the restricted maximum likelihood (ReML) method for meta-regression analysis. Subgroup analysis was used to explore sources of heterogeneity.

## 3. Results

### 3.1. Literature Search

Our search strategy identified a total of 799 related references. After deleting duplicates and non-topic-related literature, the number of papers was reduced to 316, and 130 references (omitting literature that does not include outcome measures for this study) were excluded by reading the abstract and the original text. Finally, after evaluating the full text of the remaining studies, we included 22 clinical trials (RCT+CCT). Nine studies were published in English [25,26,27,31,34,35,36,38,39], and 13 in Chinese [28,29,30,32,33,37,40,41,42,43,44,45,46], The research selection process is shown in Figure 1 and Appendix A.

### 3.2. Study Characteristics

The main features of the 22 eligible studies are shown in Table 1 and Appendix A. Articles published after 2016 accounted for 73% of these studies (16/22). The test sample size ranged from 10 to 195 (1199 total) and age ranged from 40 to 86 years. The kind of PD was mild to moderate in both Hoehn-Yahr stages. Of the 21 trials, there was only one control group, and one trial included two control groups. Therefore, the study obtained 23 independent effect sizes. In the eligible trials, there were 12, 4, and 6 studies in Tai Chi, Yoga, and Health Qigong, respectively. The control group included various interventions, such as stretching and conditioning exercises, walking, and health education. Only three studies [26,31,36] used a waitlist control to treat each exercise time for 60 min.

### 3.3. Methodological Quality

Table 2 shows the Pedro literature quality assessment scores that met study conditions. The total score ranged from 5 to 8 points. Six of the included studies were blinded and three were concealed. The dropout rates in all five studies were higher than 15%, and 9 studies used intention-to-treat analysis. For the remaining criteria of the PEDro scale, the included studies showed a high method of quality. There were no differences between the reviewers regarding the Pedro table assessment scores.

### 3.4. UPDRS (Unified Parkinson’s Disease Rating Scal) Rating Results

Fifteen articles [25,26,28,29,31,33,34,35,36,38,39,40,41,42,46] evaluated the effects of physical and mental exercise on PD motor function (Table 1). Among them, 13 articles [25,26,28,29,34,35,36,38,39,40,41,42,46] used bradykinesia items in the motor section of the Unified Parkinson’s Disease Rating Scale (UPDRSIII) to evaluate the effect of patients’ motor function, and the remaining two used UPDRS [31] and MDRSPD [33]. An asymmetrical funnel plot was presented. According to the funnel plot, two outliers were excluded (Wang et al., 2016 [46] and Liu et al., 2017 [28]). The final funnel plot is depicted in Figure 2. Heterogeneity testing of the study literature was high (*p* < 0.001, *I*^2^ = 81%), using a random-effects model for meta-analysis (Figure 3); the results showed SMD = −0.61, 95%CI (−0.95, −0.27), *p* = 0.0004 < 0.001, indicating that the physical and mental movements of Tai Chi, Yoga, and Health Qigong had significant effects on the improvement of PD motor function.

Meta-regression analysis showed that the age of publication (*p* = 0.972), sample size (*p* = 0.544), and intervention trial period (*p* = 0.582) suggested that the source of heterogeneity between studies could not be explained. Sensitivity analysis of heterogeneity and elimination revealed that Jianzhong Wang et al. [46] and Xiaolei Liu et al. [28] had a large bias in 2016, and we excluded them to conduct a meta-analysis on the remaining clinical control experiments. Heterogeneity was shown to be reduced (*I*^2^ = 53%, *p* = 0.01), SMD = −0.40, 95% CI (−0.63, −0.17), and there was significant heterogeneity between the groups (*p* < 0.001).

Subgroup analyses were performed according to characteristics of the study interventions that may cause heterogeneity differences. The results are shown in Table 3. In the gender subgroup, when the number of males tested was less than the number of females, the heterogeneity of the combined effect of the trials was significant (*I*^2^ = 92%, *p*<0.001); thus, it may be the source of differences in inter-study effect values; in the subgroup of Hoehn-Yahr disease stratification, there was a strong heterogeneity in the case of 1~2 cases (*I*^2^ = 82%, *p* < 0.01).

### 3.5. Timed Up and Go Test Score Results

Ten articles [27,28,29,32,35,36,38,39,44,45] used the timed up and go test (TUG) to assess a patient’s functional mobility. An asymmetrical funnel plot was presented. According to the funnel plot, one outlier were excluded (Guan et al., 2018 [45]). A final funnel plot is depicted in Figure 4. The heterogeneity test results of the included research literature were not significant (*p =* 0.17, *I*^2^ = 29%), so meta-analysis was performed using the fixed effect model. The results showed that total MD was significant (MD = −1.47), 95% CI (−1.80, −1.13), *p* < 0.001, indicating that Tai Chi, Yoga, and Health Qigong have a significant effect on the improvement of PD functional walking ability (Figure 5).

### 3.6. Balance Function Score Results

Twelve articles [26,29,31,33,34,39,40,41,42,44,45,46] evaluated the balance function of PD, 10 of which were evaluated by the Berg Balance Scale (BBS), 2 studies [29,34] used the Mini-Balance Evaluation Systems Test (Mini-BESTest scale) to evaluate the balance effect. A funnel chart shows no publication bias (Figure 6), a funnel plot shows no bias, and the heterogeneity test results of the included research literature was not significant (*p* = 0.29, *I*^2^ = 16%). The results showed that SMD = 0.79, 95% CI (0.62, 0.97), *p* < 0.001, the difference was significant, which indicates that Tai Chi, Yoga, and Health Qigong significantly improved PD balance function, compared with the control group (Figure 7).

### 3.7. Depression Test Score Results

Five articles used HADS [31], BDI [26], POMS [30], HAMD [46], and SCL-90 [37], and different test tools to evaluate the efficacy of depressive symptoms in PD patients. An asymmetrical funnel plot was presented. A small number of studies can lead to this result Figure 8. The heterogeneity test results of the included research literature were high (*I*^2^ = 91%, *p* < 0.01); thus, the random-effects model was used for meta-analysis. The results showed that SMD was significant (SMD = −1.61), 95% CI (−2.65, −0.57), *p* = 0.002 (Figure 9), indicating that Tai Chi, Yoga, and Health Qigong improved PD depressive symptoms better than the control group. By removing literature one by one, it is found that the research work by Jianzhong Wang et al. [46] in 2016 was highly sensitive and, thus, eliminated. The remaining clinical control experiments were analyzed by meta-analysis. The results showed that heterogeneity was reduced (*I*^2^ = 64%, *p* = 0.24), SMD = −1.14, 95% CI (−1.75, −0.54), and there were significant differences between the groups (*p* < 0.01).

Subgroup analyses were performed according to characteristics of the study interventions that may cause heterogeneity differences. The results are shown in Table 4. In the subgroup analysis, the heterogeneity between the trials with an intervention period greater than eight weeks (*I*^2^ = 92%, *p* < 0.001) and the use of adjuvant drugs in the trial also resulted in heterogeneity of the combined effects (*I*^2^ = 95%, *p* < 0.001); thus, the intervention cycle and the use of intervention drugs may be the source of the difference.

### 3.8. Quality of Life Test Score Results

A total of 6 articles that evaluated the quality of life of PD were included. Two of the literature used the PDQ-39 scale [36,43] and two used UPDRS (ADL) to assess quality of life assessment [26,38]. The remaining two articles used the ADL scale [33] and the WHO-QoL-BREF scale [37]. When the results are included, the differences in the scale of assessment effects of different scales are unified. The positive results of the results were used as the primary outcome scores for the outcome measures. The results of the experimental results with opposite outcomes were manually filled in before the data. Funnel chart showing no publication bias (Figure 10), funnel plot shows no bias, and the heterogeneity test results were included in the included studies (*p* = 0.17, *I* = 35%). Meta-analysis using a fixed-effects model, using standardized mean difference (SMD) as the combined effect scale, was done; the results showed SMD = 0.66, 95% CI (0.41, 0.91), *p* < 0.001, indicating that Tai Chi, Yoga, and Health Qigong have a significant effect on the improvement of PD quality of life (Figure 11). A funnel plot was made with pseudo 95% confidence limits.

## 4. Discussion

The clinical symptoms of PD are mainly motor related. The UPDRS Sports Subscale is the most commonly used indicator for Parkinson’s Disease motor symptom assessment. It is also a widely used clinical scale and the main indicator of motor function in many experiments. Therefore, this index was selected for analysis [47]. The meta-analysis results of Tai Chi, Yoga, and Health Qigong on motor function showed improvement (SMD = −0.61, *p* < 0.001). There is evidence that balance training in Tai Chi, Yoga, and Health Qigong can reduce the contraction of antagonist muscles [48,49,50,51]. The initial delay time of muscle activation is shortened, and reflex activity is increased. Limb tremor in Parkinson’s patients is an uncoordinated contraction of active and antagonist muscles, accompanied by continuous rhythmic contractions and relaxations. Patients can strengthen the cooperative contraction of active muscles and antagonist muscles through different forms of training. Combined with changes in movement and stimulation during movement, the number of activated transverse bridges in the muscle can be increased and muscle activity enhanced. At the same time, exercise can also strengthen the muscle and increase the muscle’s explosive force, and play a role in the rehabilitation of a patient’s motor function [52]. There is evidence that different forms of mind-body exercises can increase cerebral blood flow, improve angiogenesis, and increase brain source [53,54]. The secretion of neurotrophic factors and the activation of neuroendocrine pathways can produce an empirically dependent neuro-plastic change by stimulating brain tissue, positively affecting brain function and ontological motion perception, and promoting motor function improvement [23,25]. Because the body’s center of gravity and movements are constantly stimulating, blood flowing through the muscles of the lower limbs increases, and muscle metabolism and stress function are strengthened, stimulating muscle growth and supporting the lower limbs. At the same time, changes in movements stimulate the proprioceptive pathways, improving the body’s control and balance.

Physical and mental exercise combines the regulation of mental and respiratory states to improve the quality of life of patients [55,56]. The results of UPDRS-combined effects after different physical and mental exercises were less robust (*I*^2^ = 81%), and sensitivity and subgroup analysis were found in gender (SMD = −1.16, *p* = 0.04) and Hoehn-Yahr disease grade (SMD = −1.15, *p* = 0.002). There was heterogeneity in the subgroup; studies have found that men are more likely to develop PD and have a higher incidence [57]. Female estrogen affects the synthesis and release of dopamine and inhibits dopamine uptake, and may play a key role in the pathogenesis of Parkinson’s disease [58]. Some researchers have also pointed out that exercise performance may be influenced by gender. Women with PD are more likely to have tremors and non-motor symptoms such as tension, grief, and depression [59]. Male PD patients have lower motor function than females [60]. In addition to gender differences, Hausdorff et al. [61] also found that the motor function of PD patients is highly correlated with the severity of the disease. Although these differences exist, the overall report found that physical and mental exercise has an effect on the improvement of motor function.

In the meta-analysis of the indicators of TUG and balance function, the results showed that Tai Chi, Yoga, and Health Qigong could improve TUG (SMD = −1.47, *p* < 0.001) and balance function (SMD = 0.79, *p* < 0.001). Because the exercises of Tai Chi, Yoga, and Health Qigong include continuous movements and changes in the body’s center of gravity, the patient’s lower extremity muscle strength is strengthened, and proprioceptive perception work is strengthened, which improves the ability to control balance. In addition to the performance of motor symptoms, about 50% of patients with Parkinson’s Disease will suffer from a variety of non-motor disorders. Depression is a common comorbidity. Tai Chi, Yoga, and Health Qigong can significantly reduce depressive symptoms (SMD = −1.61, *p* = 0.002). This effect, in part, may occur through the expression of neurotrophic substances and the synthesis and expression of induced monoamine neurotransmitters, which can reduce depressive symptoms [62]. We demonstrated that interventions may be the source of heterogeneity, and the use of clinical antidepressants and drugs for the prevention and treatment of other common diseases may increase the effect of trial results. The trial intervention cycle also affects assessment outcomes. Studies have shown that exercise contributes to the improvement of depressive symptoms, and long-term trials may have a more favorable outcome [63,64]. However, some researchers have pointed out that, in addition to the test cycle, exercise intensity and exercise frequency also have an impact. A meta-analysis of a prospective study confirmed that early moderate-to-intense activity can reduce the risk of Parkinson’s Disease [65], and many concurrent diseases (diabetes, hypertension, hyperlipidemia, obesity, and osteoporosis) can benefit from strenuous exercise [66,67].

Tools commonly used in the trial to assess patients’ quality of life included PDQ-39 and UPDRS Part II Quality of Life Assessment (ADL) scales. Meta-analysis showed a significant improvement in quality of life (SMD = 0.66, *p* < 0.001). Because PD patients have motor and non-motor dysfunction, which leads to a significant decline in quality of life, the negative effects of non-motor dysfunction are especially more serious. Physical and mental methods such as Tai Chi and Health Qigong can be used as medical aids to improve the clinical symptoms of patients. It has been proven that these methods can reduce the negative emotions brought about by the disease, increase subjective happiness, improve quality of life, and promote rehabilitation. In summary, Tai Chi, Yoga, and Health Qigong improve motor function, reduce depressive symptoms, and increase quality of life of PD patients.

This study has several limitations: First, Tai Chi, Yoga, and Health Qigong have many different styles and characteristics. Many of the PD training programs included in this study were developed specifically for research purposes. The differences between these programs will also cause variations between studies. Second, because there are relatively few studies available to evaluate most of the results, the findings of this study need to be validated with future trials. Third, we did not distinguish between the results of pre-and-post medication assessments as traditional methods did. Further physical and mental exercise trials targeting specific PD subgroups or stratification will make a significant contribution to the current evidence base. Finally, there is a big discrepancy between the number of studies published in English and Chinese. This is because Tai Chi, Yoga, and Health Qigong are more popular among Asian populations and recognized by Asian medicine [68,69]. This knowledge is not popular among non-Chinese people, and it is difficult to assess the impact of this discrepancy.

## 5. Conclusions

First, in summary, Tai Chi, Yoga, and Health Qigong improve motor function, reduce depressive symptoms, and increase quality of life for PD patients. These findings have important implications for clinical practice, underscoring the need to promote alternative forms of exercise for PD patients. In addition, there is heterogeneity in the effect of exercise symptoms and depression indicators. It is found that the cause of heterogeneity of motor symptoms may be related to patient gender, Hoehn-Yahr disease classification, and the cause of heterogeneity of depressive symptoms may be related to the trial intervention period. Intervention measures are relevant in this research. Due to the limited number and quality of the literature, rigorous trials are needed to better describe the role of physical and mental exercise in PD, and to guide individuals with different disease subtypes and symptom burdens to choose the optimal dose and specific protocols for a more scientific intervention in the PD symptom procedure.

## Figures and Tables

**Figure 1 ijerph-17-00031-f001:**
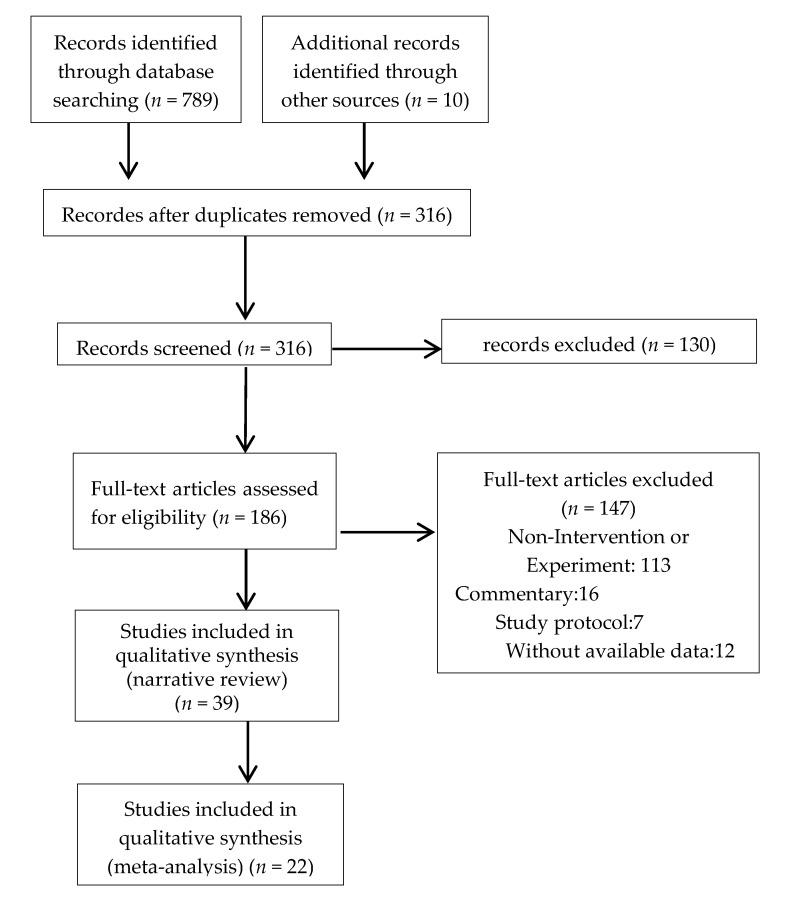
Flowchart displaying study selection.

**Figure 2 ijerph-17-00031-f002:**
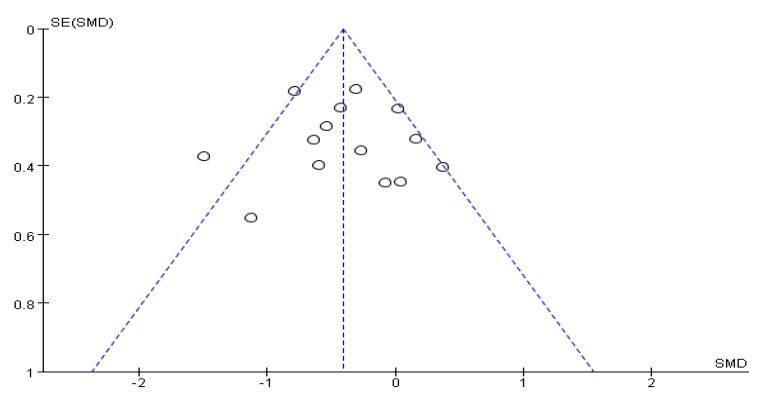
Funnel plot for UPDRS rating results after removing outliers.

**Figure 3 ijerph-17-00031-f003:**
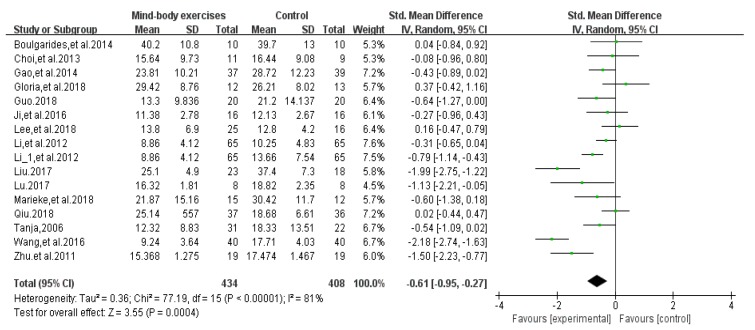
Effects of mind-body exercises on UPDRS Rating Results.

**Figure 4 ijerph-17-00031-f004:**
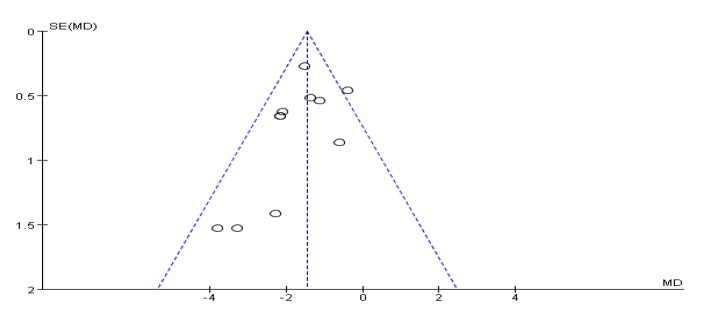
Funnel plot for TUG test score results after removing outliers.

**Figure 5 ijerph-17-00031-f005:**
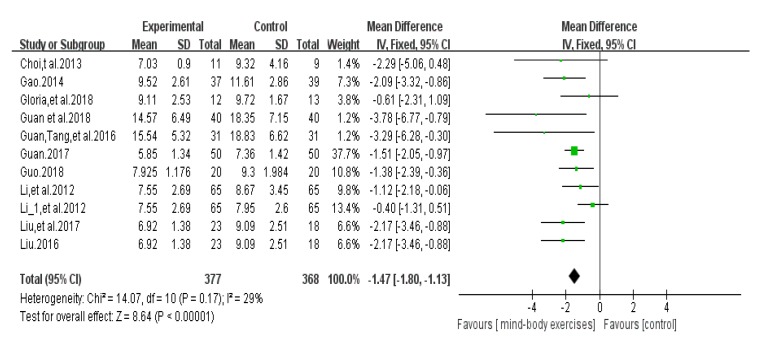
Effects of mind-body exercises on timed up and go test score results.

**Figure 6 ijerph-17-00031-f006:**
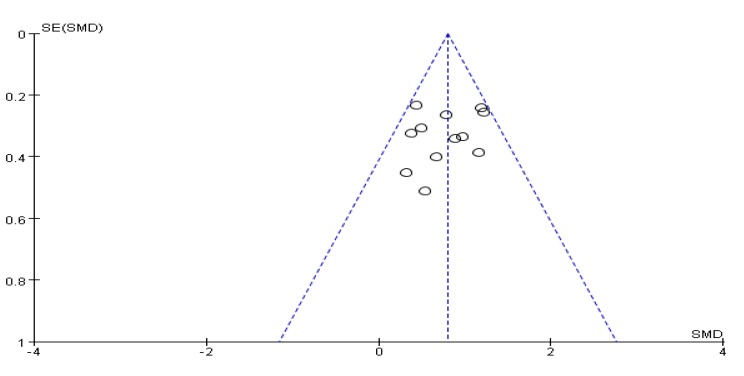
Funnel plot for balance function score results after removing outliers.

**Figure 7 ijerph-17-00031-f007:**
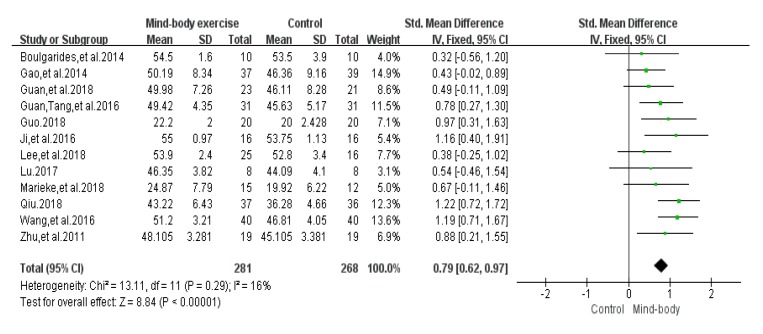
Effects of mind-body exercises on balance function score results.

**Figure 8 ijerph-17-00031-f008:**
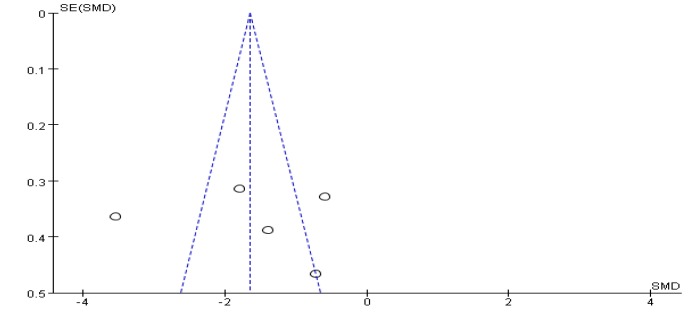
Funnel plot for depression test score results after removing outliers.

**Figure 9 ijerph-17-00031-f009:**
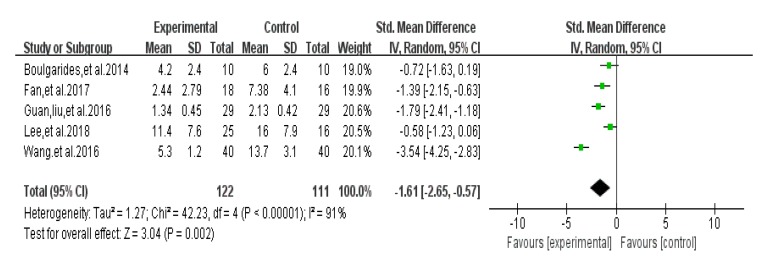
Effects of mind-body exercises on depression test score results.

**Figure 10 ijerph-17-00031-f010:**
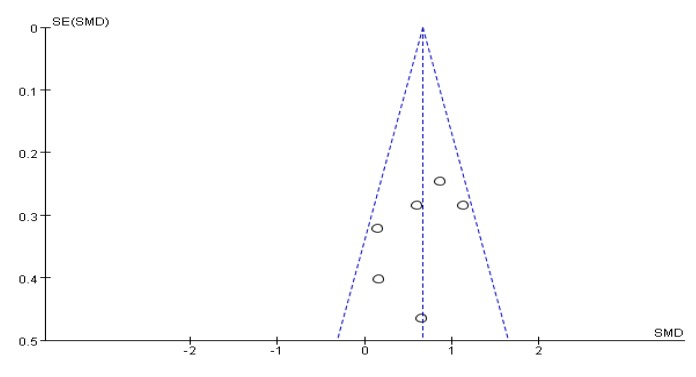
Funnel plot for quality of life test score results after removing outliers.

**Figure 11 ijerph-17-00031-f011:**
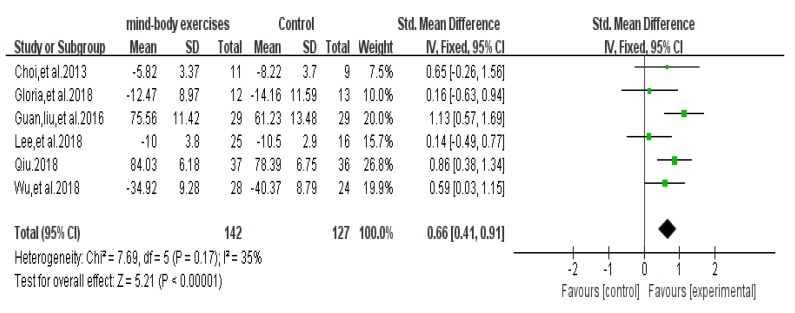
Effects of mind-body exercises on quality of life test score results.

**Table 1 ijerph-17-00031-t001:** Summary of included studies.

First Author, Year, Country	Sample Size	Age (Rang)	Treatment Duration (Week)	Experimental Group Intervention	Control Group Intervention	Main Outcome Assessments
Marieke, et al. (2018) US	30 (10%)	59~75	8	YG:2/week Yoga	CG:usual care	UPDRS/Mini-BEST
Qiu (2018) China	80 (8.8%)	59~71	24	YG:(60 min Yoga + 30 min bicycle) × (2~3)/week + UC	CG:40–60 min/day + usual care	UPDRS/ADL/BBS/MDRSPD
Guan (2017) China	100	65~74	24	YG:Yoga + UC	CG:usual care	TUGT
Boulgarides, et al. (2014) US	10	43~77	8	YG:60 min/week (Yoga)	CG:wait-list	HADS/UPDRS/BBS
Lee, et al. (2018) Korea	32 (18.8%)	58~73	8	QG:2 × 60 min/week (Qigong + dance)	CG:waiting-list	UPDRS/BBS/BDI/PDQL
Tanja, et al. (2006) German	56 (12.5%)	56~71	8	QG:60 min/week Qigong + Med	CG:Med	UPDRSIII
Liu (2016) China	54 (24%)	59~72	10	QG:5 × 60 min/week Qigong + Med	CG:Med	TUG
Liu, et al. (2017) China	52 (21.2%)	50~65	10	QG:5 × 60 min/week Qigong + Med	CG:Med	UPDRS III/TUG
Guo (2018) China	40	57~71	12	QG:2 × (60~80) min/week Qigong	CG:usual care	UPDRSIII/TUGT/Mini-BEST
Fan, et al. (2017) China	36 (5.6%)	42~86	8	QG:5 × 60 min/ week Qigong	CG:usual care	POMS
Li, et al (2012) US	195 (9.7%)	40~85	24	TC:2 × 60 min/week Qigong	CG1:Resistance CG2:Stretching	UPDRSⅢ/TUG
Gloria, et al. (2018) US	32 (15.6%)	40~75	24	TC:2 × 60/week Tai Chi	CG:usual care	UPDRS/TUG/PDQ-39
Guan, Liu, et al. (2016) China	58	64~75	12	TC:4 × 60 min/week Tai Chi	CG:usual care	SCL-90/ WHO-QoL-BREF
Choi, et al. (2013) Korea	22 (9%)	53~73	12	TC:3 × 60 min/week Tai Chi	CG:non-exercise	UPDRS/ADL/TUG
Gao, et al. (2014) China	40 (5%)	59~78	12	TC:3 × 60 min/week Tai Chi + Med	CG:Med	UPDRS/TUGT/ BBS
Lu (2017) China	16	60~76	8	TC:5 × (40~60) min/week TaiChi + Med	CG:Med	UPDRSⅢ/BBS
Ji, et al. (2016) China	38 (15.8%)	44~71	12	TC:60 min/day Tai Chi + Med	CG:Med	UPDRSⅢ/BBS
Zhu, et al. (2011) China	40 (5%)	52~73	4	TC:5 day × (30~40) ×2 Tai Chi + Med	CG:Med+(40~60) minx5day	UPDRSⅢ/BBS
Wu (2018) China	55 (5.5%)	66~70	16	TC:4 × 40 min/week Tai Chi + Med	CG:Med+ usually exercise	PDQ-39
Guan, Tang, et al. (2016) China	62	65~76	12	TC:4 × 60 min/week Tai Chi + UE	CG:usual exercise	TUGT/BBS
Guan, et al. (2018) China	81 (1.3%)	62~75	24	TC:4 × 60 min/week Tai Chi + UE	CG:Usual Care	TUGT/BBS
Wang, et al. (2016) China	80	55~75	16	TC:2 × (50~60) min/day Tai Chi + Med	CG:Med	UPDRSⅢ/BBS/HAMD

Note: TC = Tai chi; YG = Yoga ; QG = qigong; Med = medicine; UC = usual care; UE = usually exercise; CG = control group; UPDRS = Unified Parkinson’s Disease Rating Scale; BDI = Beck Depression Inventory; HAMD = Hamilton Depression Rating; SCL-90 = Self-reporting Inventory-90; BBS = Berg Balance Scale; PDQ(L)-39 = The 39-Item Parkinson’s Disease Questionnaire; TGU/TUGT = The Timed Up and Go Test; Mini-BEST = Mini-Balance Evaluation; MDRSPD = Motor Dysfunction Rating Scale for Parkinson’s Disease; ADL = Activities of Daily Living; HADS = Hospital Anxiety and Depression Scale; WHO-QoL-BREF = The World Health Organization Quality of Life (WHOQOL)-BREF; POMS = Profile of Mood States.

**Table 2 ijerph-17-00031-t002:** Study quality assessment of all selected trials.

Study	Item 1	Item 2	Item 3	Item 4	Item 5	Item 6	Item 7	Item 8	Item 9	Item 10	Item 11	Total
Marieke, et al. (2018)	1	1	0	1	0	0	1	1	0	1	1	7
Qiu (2018)	1	1	0	1	0	0	0	1	1	1	1	7
Guan (2017)	1	1	0	1	0	0	0	1	0	1	1	6
Boulgarides (2014)	1	1	0	1	0	0	0	1	1	1	1	7
Lee et al. (2018)	1	1	1	1	0	0	1	0	0	1	1	7
Tanja et al. (2006)	1	1	1	1	0	0	0	1	1	1	1	8
Liu (2016)	1	1	0	1	0	0	0	0	0	1	1	5
Liu (2017)	1	1	0	1	0	0	0	0	0	1	1	5
Guo (2018)	1	0	0	1	0	0	0	1	1	1	1	6
Fan et al. (2017)	1	1	0	1	0	0	0	1	0	1	1	6
Li et al. (2012)	1	1	1	1	0	0	1	1	0	1	1	8
Gloria, et al. (2018)	1	1	0	1	0	0	1	0	1	1	1	7
Guan, Liu et al. (2016)	1	1	0	1	0	0	0	1	1	1	1	7
Choi et al. (2013)	1	1	0	1	0	0	1	1	0	1	1	7
Gao et al. (2014)	1	1	0	1	0	0	0	1	0	1	1	6
Lu (2017)	1	1	0	1	0	0	0	1	1	1	1	7
Ji et al. (2016)	1	1	0	1	0	0	0	0	0	1	1	5
Zhu et al. (2011)	1	1	0	1	0	0	1	1	0	1	1	7
Wu. (2018)	1	1	0	1	0	0	0	1	0	1	1	6
Guan, Tang, et al. (2016)	1	1	0	1	0	0	0	1	1	1	1	7
Guan et al. (2018)	1	1	0	1	0	0	0	1	0	1	1	6
Wang et al. (2016)	1	1	0	1	0	0	0	1	1	1	1	7

Note: 0: does not meet the criteria; 1: meets the criteria. Criteria (without eligibility criteria) were used to calculate the total PEDro score; Item 1 = Eligibility criteria; Item 2 = Random sequence; Item 3 = Allocation concealment; Item 4 = Similar at baseline; Item 5 = Subjects blinded; Item 6 = Therapists blinded; Item 7 = Assessors blinded; Item 8 = < 15% dropouts; Item 9 = Intention-totreat analysis; Item 10 = Between-group comparisons; Item 11 = Point measures and variability data.

**Table 3 ijerph-17-00031-t003:** Subgroup analysis of differences between experimental and control groups in UPDRS rating results.

Figure	N		SMD (95% CI)	*I*^2^ (%)	*p*-Value	(SMD) *p*-Value
Gender						
Male > Female	10		−0.59 (−0.81,−0.37)	37%	0.12	0.000
Male = Female	1		0.37 (−0.42,1.16)	—	—	0.000
Male < Female	4		−1.16 (−2.28,−0.03)	92%	0.000	0.04
Hoehn-Yahr						
1–2	5		−1.15 (−1.89,−0.41)	82%	0.000	0.002
2–3	4		−0.55 (−1.60,0.49)	85%	0.000	0.30

Note: “—” Heterogeneity test cannot be conducted due to the lack of literature; N is the required literature statistics. The number of people in Hoehn-Yahr grades 1–3 and 1–4 were not included in the subgroup analysis.

**Table 4 ijerph-17-00031-t004:** Subgroup analysis of differences between experimental and control groups in depression test score results.

Feature	Number	SMD (95% CI)	*I*^2^ (%)	*p*-Value	(SMD) *p*-Value
Invention	Time				
≤ 8	3	−0.88 (−1.39, −0.38)	25%	0.26	0.0006
> 8	2	−2.66 (−4.37, −0.94)	92%	0.0003	0.002
Invention	medicine				
Report	3	−1.96 (−3.58, −0.35)	95%	0.000	0.02
Non-Report	2	−1.10 (−1.76, −0.45)	19%	0.27	0.0009

Note: Report = there have been reports of the use of specific drugs, Non-Report is not reported.

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
