# Peer review of "The Impact of Mind-Body Exercises on Motor Function, Depressive Symptoms, and Quality of Life in Parkinson’s Disease: A Systematic Review and Meta-Analysis"

_ijerph, 2019, doi:10.3390/ijerph17010031_

Round 1

Reviewer 1 Report

I am not a native speaker of English, but there are many errors of the language which makes the following of the text difficult. They must be improved prior to the publication of the study.

Page 4, line 139 - why 130 studies were exluded after reading abstracts and full text? Some justificiation should be given.

There is a big discrepacy between the number studies published in English and Chinese. This can be explained by the fact, that tai chi, yoga and quigong are more popular among asian populations, and are recognized by asian medicine. The knowlegde of Chinese is not common among non-Chinese natives it is difficult to assess the impact of this discrepancy.

The full names of all the tests should be given, either when they first appear in the paper, or in a small dictionary as the Appendix.

Page 9, line 73 - First sentence - this is true for the second phase of PD. In the third phase psyochotic symptoms, together with hallucinations, memory loss, distrorted time perceptions are accompanying the motor disorders. I presume that in most of the studies the patients were in the second phase.

Author Response

Response to Reviewer 1 Comments

Point 1: I am not a native speaker of English, but there are many errors of the language which makes the following of the text difficult. They must be improved prior to the publication of the study. 

Response 1: I have been actively revising and asked native English speakers to check it carefully for me. I will upload it again after the modification.

Point 2: Page 4, line 139 - why 130 studies were exluded after reading abstracts and full text? Some justificiation should be given.

Response 2: The corresponding supplementary explanations have been made in this place, exclude literature that does not include outcome measures for this study.

Point 3: There is a big discrepacy between the number studies published in English and Chinese. This can be explained by the fact, that tai chi, yoga and quigong are more popular among asian populations, and are recognized by asian medicine. The knowlegde of Chinese is not common among non-Chinese natives it is difficult to assess the impact of this discrepancy.

Response 3: I have explained it in the discussion section of the article.

Point 4: The full names of all the tests should be given, either when they first appear in the paper, or in a small dictionary as the Appendix.

Response 4: Okay, it has been modified accordingly. I supplemented where the full name appeared

Point 5: Page 9, line 73 - First sentence - this is true for the second phase of PD. In the third phase psyochotic symptoms, together with hallucinations, memory loss, distrorted time perceptions are accompanying the motor disorders. I presume that in most of the studies the patients were in the second phase.

Response 5: Yes, this is just one of my declarative statements. As you said, this is just a pavement for the discussion below.

Reviewer 2 Report

The authors conducted a systematic review and meta-analysis to investigate the impact of mind-body exercises on motor function, depressive symptoms and quality of life in patients with Parkinson's disease.

The systematic search of the literature appears to be well conducted as it involved a high number of databases and a good search strategy. Additionally, the authors evaluated the quality of included studies. The analyses appear to be sound.

The authors should add an evaluation of the publication bias. 

English grammar and language should be carefully revised by a native language speaker as a high number of sentences are either incomplete or contain errors that make the manuscript difficult to read.

In the material and methods section, at line 93, the authors should clarify the meaning of the sentence "We also manually determined relevant studies."

In the introduction, line 54, "substantia" should be replaced with "substantia nigra".

The sentence starting with "some options..." at line 69 is incomplete.

In the abstract, in the expression "Parkinson'spatients" a space is missing.

At line 25 the word "data" should be replaced with "date".

At line 28: the expression "this studies" should be replaced with "these studies".

Author Response

Response to Reviewer 2 Comments

Point 1: The authors should add an evaluation of the publication bias.

Response 1: I have added a publication bias assessment.

Point 2: English grammar and language should be carefully revised by a native language speaker as a high number of sentences are either incomplete or contain errors that make the manuscript difficult to read.

Response 2: I have been actively revising and asked native English speakers to check it carefully for me. I will upload it again after the modification.

Point 3: In the material and methods section, at line 93, the authors should clarify the meaning of the sentence "We also manually determined relevant studies."

Response 3: It has been revised to "We also emailed the original text and downloaded data from supplementary source sites.”

Point 4: In the introduction, line 54, "substantia" should be replaced with "substantia nigra".

Response 4: This has been changed to "substantia  nigra" based on the modification requirements.

Point 5: The sentence starting with "some options..." at line 69 is incomplete.

Response 5: It has been added here in its original meaning. It has been modified to“Based on the above factors, it is urgent to find other effective PD treatment programs. Although the occurrence of neurodegenerative changes can not be prevented, but exercise can be used to assist the rehabilitation of PD patients,Studies have reported that  dance [11], resistance training [12] and stretching exercise [13] are effective in improving the function of patients with Parkinson's disease” .

Point 6: In the abstract, in the expression "Parkinson'spatients" a space is missing.

Response 6: The space here has been filled in.

Point 7: At line 25 the word "data" should be replaced with "date".

Response 7: This has been changed to "date" based on the modification requirements.

Point 8: At line 28: the expression "this studies" should be replaced with "these studies".

Response 8: This has been changed to "these studies" based on the modification requirements.

Reviewer 3 Report

General Comments

Due to the highest quality evidence of systematic reviews and meta-analyses, the authors need to provide the PRISMA statement. I have detected some grammatical errors, and please have a final read-through manuscript. The Authors must improve the discussion and provide the appropriate references. This is an important weaknesses of the manuscript. Authors need to recheck all figure and table titles. There were many mistakes.

Abstract

The“including 23 evaluated articles. The Pedro quality score of 6 points or more occured for 87% (20/23) of this studies, of which 22 were randomized controlled trials with a total of 1259 subjects, and the trial intervention time ranged from 4 to 24 weeks.”should be described in the result. (lines 27-30, page 1)

Introduction

“Depression is the most common comorbidity of PD [5]; drugs can not solve non-motor conditions such as depression and anxiety [6,7] (lines 59-60, page 2)” is too arbitrary “Such options dance [11], resistance training [12] and stretching exercise [13].” There is a obvious grammatical error (lines 69, page 2).

Methods

Why authors enrolled the controlled clinical trial (CCT)? Did authors detect the potential publication bias ? (the funnel plot)

Result

“The results showed that SMD was significant (SMD=0.79), 95% CI (0.62, 0.97), P<0.001, indicating Tai Chi, The improvement effect of yoga and Qigong on PD balance function was significantly improved compared with the control group (Figure 4).”(lines 228-231, page 7-8) There is a obvious grammatical error. Authors need to recheck all figure and table titles. There were many mistakes. Should discuss Tai Chi, yoga and Qigong, separately Is there any variabilities test for difference between Control group intervention and Experimental group intervention?

Discussion

Authors need to provide the reference that why improved motor function according UPDRS score can explain“There is evidence that balance training in Tai Chi, Yoga, and Qigong can reduce the contraction of antagonist muscles. The initial delay time of muscle activation is shortened, and the reflex activity is increased.”(lines 278-279, page9)

2.“There is also evidence that different forms of mind-body exercises can increase cerebral blood flow, improve angiogenesis, increase brain source. The secretion of neurotrophic factors and the activation of neuroendocrine pathways can produce an empirically dependent neuro-plastic change by stimulating brain tissue, positively affecting brain function and ontological motion perception, and promoting motor function improvement.”There seems not to be mentioned in the reference article.(lines 281-285, page9)

3.“In the meta-analysis of two other indicators of exercise symptoms, timed up and go test (TUG), the results showed that Tai Chi, Yoga, and Health Qigong were in TUG (SMD=-1.47, P<0.001). Improvements in the balance function (SMD=0.79, P<0.001) are improved.”(lines 299-301, page10) Here are some mistakes. It is difficult to be read.

4.“Tai Chi, Yoga, and Health Qigong may strengthen the muscle of the lower limbs and strengthen the body's sensory function, which in turn may improve the ability of body control and balance.”Authors need to provide the reference.

5.“This effect, in part, may occur through the expression of neurotrophic substances and the synthesis and expression of induced monoamine neurotransmitters, which can reduced depressive symptoms[62]”(lines 319-320, page10) This part should be in previous paragraph. And please explain the relationship between the Mind-body Exercises and the quality of life in PD patients.

Author Response

Response to Reviewer 3 Comments

Point 1: Due to the highest quality evidence of systematic reviews and meta-analyses, the authors need to provide the PRISMA statement. I have detected some grammatical errors, and please have a final read-through manuscript. The Authors must improve the discussion and provide the appropriate references. This is an important weaknesses of the manuscript. Authors need to recheck all figure and table titles. There were many mistakes. 

Response 1: Recommendation received, article PRISMA statement has been supplemented. I am carefully revising the references in the discussion section and checking the titles of the charts

Point 2:  The“including 23 evaluated articles. The Pedro quality score of 6 points or more occured for 87% (20/23) of this studies, of which 22 were randomized controlled trials with a total of 1259 subjects, and the trial intervention time ranged from 4 to 24 weeks.”should be described in the result. (lines 27-30, page 1)

Response 2: I think this is just a description of the method, and it cannot be used as the result data of my research. So it should be more suitable in the method description.and I confirm that I made a change here.

Point 3: “Depression is the most common comorbidity of PD [5]; drugs can not solve non-motor conditions such as depression and anxiety [6,7] (lines 59-60, page 2)” is too arbitrary “Such options dance [11], resistance training [12] and stretching exercise [13].” There is a obvious grammatical error (lines 69, page 2).

Response 3: This place is not accurate enough and has been modified(page 2).

Point 4: Why authors enrolled the controlled clinical trial (CCT)? Did authors detect the potential publication bias ? (the funnel plot)

Response 4: Yes, referring to the existing literature, it is suggested that the CCT test can also be included in the meta-analysis. This article also made a publication bias (added in the text)and found that it did not cause a difference.

Point 5: “The results showed that SMD was significant (SMD=0.79), 95% CI (0.62, 0.97), P<0.001, indicating Tai Chi, The improvement effect of yoga and Qigong on PD balance function was significantly improved compared with the control group (Figure 4).”(lines 228-231, page 7-8) There is a obvious grammatical error. Authors need to recheck all figure and table titles. There were many mistakes. Should discuss Tai Chi, yoga and Qigong, separately Is there any variabilities test for difference between Control group intervention and Experimental group intervention?

Response 5: This place has been modified for expression errors and title issues. Due to the limited number of studies that meet the criteria, a large error will occur if they are discussed separately, which is also the deficiency of this study.

Point 6: Authors need to provide the reference that why improved motor function according UPDRS score can explain“There is evidence that balance training in Tai Chi, Yoga, and Qigong can reduce the contraction of antagonist muscles. The initial delay time of muscle activation is shortened, and the reflex activity is increased.”(lines 278-279, page9)

Response 6: Limb tremor in Parkinson's patients is an uncoordinated contraction of active and antagonist muscles accompanied by continuous rhythmic contractions and relaxations. Patients can strengthen the cooperative contraction of active muscles and antagonist muscles through different forms of training. Combined with changes in movement and stimulation during movement, the number of activated transverse bridges in the muscle can be increased and muscle activity can be enhanced.

Point 7: “There is also evidence that different forms of mind-body exercises can increase cerebral blood flow, improve angiogenesis, increase brain source. The secretion of neurotrophic factors and the activation of neuroendocrine pathways can produce an empirically dependent neuro-plastic change by stimulating brain tissue, positively affecting brain function and ontological motion perception, and promoting motor function improvement.”There seems not to be mentioned in the reference article.(lines 281-285, page9)

Response 7: The pertinence of this reference may not be straightforward and has been modified to[23]igmond, M.J., Cameron, J.L., et al.Neurorestoration by physical exercise: moving forward. Parkinsonism and Related Disorders 18 (Suppl 1). 2012,S147–150.

Point 8: “In the meta-analysis of two other indicators of exercise symptoms, timed up and go test (TUG), the results showed that Tai Chi, Yoga, and Health Qigong were in TUG (SMD=-1.47, P<0.001). Improvements in the balance function (SMD=0.79, P<0.001) are improved.”(lines 299-301, page10) Here are some mistakes. It is difficult to be read.

Response 8: The statement here is modified to “In the meta-analysis of the indicators of TUG and balance function,the results show that Tai Chi, Yoga, and Health Qigong could improve on TUG (SMD = -1.47, P <0.001) and balance function (SMD = 0.79, P <0.001). Because the exercises of Tai Chi, Yoga, and Health Qigong include continuous movements and changes in the body's center of gravity, the patient's lower extremity muscle strength is strengthened, and proprioceptive perception work is strengthened, which improves the ability to control balance”.

Point 9: “Tai Chi, Yoga, and Health Qigong may strengthen the muscle of the lower limbs and strengthen the body's sensory function, which in turn may improve the ability of body control and balance.”Authors need to provide the reference.

Response 9: Because the body's center of gravity and movements are constantly stimulating, blood flowing through the muscles of the lower limbs increases, muscle metabolism and stress function strengthen, which will stimulate muscle growth and strengthen the lower limbs. At the same time, changes in movements will stimulate the proprioceptive pathways It is strengthened so that the body's ability to control and balance can be improved.

Point 10: “This effect, in part, may occur through the expression of neurotrophic substances and the synthesis and expression of induced monoamine neurotransmitters, which can reduced depressive symptoms[62]”(lines 319-320, page10) This part should be in previous paragraph. And please explain the relationship between the Mind-body Exercises and the quality of life in PD patients.

Response 10: I have adjusted the part according to the proposed changes.

Because PD patients have motor and non-motor dysfunction, which leads to a significant decline in quality of life, especially the negative effects of non-motor dysfunction are more serious. The physical and mental methods such as Tai Chi and Qigong can be used as medical aids to improve the clinical symptoms of patients. It has been proven that it can reduce the negative emotions brought by the disease, improve the subjective happiness and improve the quality of life.f patients with Parkinson's and promote the rehabilitation of patients, and improve their quality of life.